# Spatio-temporal Diffusion Transformer for Action Recognition

## Abstract

Video action recognition has aroused the research interest of many scholars, and has been widely applied in public surveillance, video review, sports events and other fields. However, the high similarity of video backgrounds and the long time span of action bring serious challenges to action recognition. In this work, we propose a spatio-temporal diffusion transformer (STD-Former) to improve the recognition accuracy of long-distance and fine-grained actions from redundant backgrounds. STD-Former utilizes a two-branch network to extract the spatiotemporal and temporal information of video respectively. First, we present a parallel transformer module to capture the spatiotemporal feature of actions through attention mechanism and convolutional structure in the spatiotemporal branch. Secondly, a cross transformer module integrating the feature of spatiotemporal branch is constructed to explore the long-distance temporal dependency relationship of actions in the temporal branch. In addition, inspires by the advantage of the diffusion principle in exploring long-term temporal dependency, we design a novel plug-and-play spatiotemporal diffusion module that feeds back the feature extracted from the temporal branch to the spatiotemporal branch, thereby enhancing the ability of model to capture motion information from a large number of redundant backgrounds. Finally, in order to learn the fine-grained action information between adjacent video sequences, another plug-and-play significant motion excitation module is established to convert the spatial information of adjacent video frames into the motion feature. The experimental results on Something Something V1 and V2 datasets demonstrate that STD-Former can more accurately identify the fine-grained action and has favorable robustness than the current state-of-the-art action recognition models.

## 1 Introduction

Action recognition can efficiently identify and analyze video actions (Sun et al., 2023b), and has been applied in public surveillance (Elharrouss et al., 2021), medical monitoring (Hang & Li, 2023), video review (Sun et al., 2023b), sports events (Tong et al., 2022) and so on. However, it still faces multiple challenges (Ramanathan et al., 2014). Different actions have considerable temporal variance, which struggles to extract short-term motion cues and characterize the action over long time spans. Meanwhile, the intra-class differences and inter-class similarities (Akila & Chitrakala, 2019) are significant for some video actions. The forms of actions with same category are various under different circumstances (for instance, running). Certain actions, such as striding and walking, are highly similar in their representation, making them indistinguishable according to spatial configuration and motion characteristics.

In recent years, deep learning based action recognition methods are constantly emerging, which could be classified into the convolutional neural network (CNN) and Transformer based models. The CNN-based action recognition models mainly included three types of architectures: 2D CNN, 3D CNN, and two-stream network. The 2D CNN-based recognition methods employed two-dimensional convolution to capture action appearance information and extract spatial and temporal features of video, such as temporal relation network (TRN) (Zhou et al., 2017), temporal shift module (TSM) (Lin et al., 2019), etc. Given that videos encompass not only static appearance features but also motion and temporal features, 2D CNN-based methods may not fully capture the long-term temporal relationship of video actions. Hence, many researchers utilized 3D convolution to directly

extract spatiotemporal feature and capture the contextual information of video actions, for instance, C3D (Tran et al., 2015), I3D (Carreira & Zisserman, 2017), R(2+1)D (Tran et al., 2018), etc. Although the 3D CNN-based method is capable of extracting spatiotemporal feature of video, they usually have a large number of parameters, so their training speed is slower than that of the 2D CNN-based methods.

In addition, a pioneering two-stream approach proposed by Simonyan & Zisserman (2014) employed two distinct models to process RGB and optical flow information concurrently. Subsequently, some researchers (Feichtenhofer et al., 2016; 2017; Wu et al., 2018) paid attention to the design strategy of two-stream network. The existing two-stream methods depended on dense sampling of video and pre-extracted optical flow feature, which needed substantial storage and computational resource. At the same time, although the two-stream network excels at capturing short-term motion feature, the long-term action feature is also crucial for accurate action recognition.

Transformer (Vaswani et al., 2017) focuses on global feature of sample through its self-attention mechanism. Therefore, scholars leveraged Transformer to model long-distance action dependency relationships and proposed many action recognition methods based on transformer (Fan et al., 2021; Yan et al., 2022; Yang et al., 2023). A convolution-free spatiotemporal transformer method (Bertasius et al., 2021) gave a new paradigm for transformer architecture, which first integrated temporal and spatial attention. Subsequently, various video feature fusion strategies (Shah et al., 2024) are presented to improve the performance of model. Motion-Former (Patrick et al., 2021) integrated trajectory attention and aggregated the spatial and temporal features along implicit motion paths. Li et al. (2023) proposed an improved visual transformer model, which improved spatiotemporal attention mechanism and temporal dependency of actions. A convolutional transformer architecture (Wu et al., 2021a) was proposed by embedding convolution into the self-attention and introducing a compressed projection to enhance local information representation. Convolutional projection (Yuan et al., 2021) was designed to accurately extract the feature of image patches through hierarchical category token attention. The ability of transformer to capture global information could improve the model performance for the long video sequences, but the inherent background redundancy in video would lead to insufficient capture of fine-grained features in the transformer-based action recognition model.

In this paper, we propose a spatio-temporal diffusion transformer (STD-Former) to accurately identify fine-grained actions with long time spans. STD-Former contains two branches with the transformer architecture: a spatiotemporal branch and a temporal branch, which extract the spatiotemporal features of video and the motion features of the moving subject, respectively. In the spatiotemporal branch, we construct a parallel transformer module to extract global spatiotemporal features and enhance local temporal information via a two-dimensional convolutional structure. For the temporal branch, multiple cross transformer modules are utilized to integrate the features from both branches and model long-range temporal dependency. Furthermore, we present a spatiotemporal diffusion module, which continuously feeds back the temporal information from temporal branch to the spatiotemporal branch, thus strengthening the spatiotemporal features. Additionally, a motion excitation module is designed to extract the moving information from adjacent video sequences and embedded into two branches. To verify the performance of STD-Former, we conducted a series of experiments on the Something Something V1 and V2 datasets. The experimental results demonstrate that STD-Former achieves higher accuracy than most mainstream models. The main contributions of this paper are as follows:

(1) In this paper, we propose a spatio-temporal diffusion transformer (STD-Former) with a spatiotemporal branch and a temporal branch for action recognition. The two branches respectively consist of multiple parallel transformer modules and cross transformer modules to extract global spatiotemporal features of video and the temporal information of actions.

(2) We devise a spatiotemporal diffusion module to explore the long-term dependency of video actions through the temporal information propagation between the two branches.

(3) To effectively capture detail information of fine-grained actions, a lightweight salient motion excitation module is integrated into both branches of STD-Former, which extracts key motion features from adjacent video sequences.

## 2 RELATED WORK

### 2.1 CNN-BASED ACTION RECOGNITION METHODS

At present, the architectures of the backbone for the existing CNN-based action recognition models mainly include 2D CNN, 3D CNN, and two-stream networks. The early action recognition methods (Yuan et al., 2022; Sudhakaran et al., 2023; Qiu et al., 2017) were based on 2D convolution. Li et al. (2020) introduced a novel temporal excitation and aggregation (TEA) approach to effectively capture short-range and long-range temporal dynamics in videos. MSNet(Kwon et al., 2020) extracted displacement tensors from adjacent video sequences for motion representation learning. Wang et al. (2021) presented a novel video architecture, temporal difference network (TDN), to capture multi-scale temporal information for efficient action recognition by using a temporal difference operator. Li et al. (2021) proposed a video classification model called CT-Net, which employed channel tensorization to enhance feature interaction and receptive fields for improving classification accuracy. Multi-view fusion network (MVFNet) (Wu et al., 2021b) modeled multi-view features of video, thereby enhancing the recognition performance of model. Temporal adaptive module (TAM) (Liu et al., 2021) employed an attention mechanism to fuse multi-scale features. Wang et al. (2023a) introduced an adjoint enhancement network (AE-Net), which addressed the challenges of motion information loss and misalignment of temporal attention through global adjoint enhancement module. Peng & Tseng (2023) presented a multi-scale motion-aware (MSMA) module to effectively capture motion information at different scales. 2D CNN-based action recognition methods could efficiently extract the spatial feature, but they were limited in acquiring temporal dynamics, particularly long-term dependency.

In contrast, 3D CNN-based action recognition models can extract spatiotemporal features of video directly, capturing video action patterns effectively. Considering the high computational cost of 3D convolution, pseudo 3D (P3D) (Qiu et al., 2017) and spatiotemporal 3D (S3D) (Xie et al., 2018) decomposed 3D convolution into a combination of 2D and 1D convolutions to extract spatiotemporal information. However, 3D CNN-based methods still have a large number of parameters, and the training speed of models is slow.

Additionally, some two-stream based action recognition networks independently processed RGB and optical flow information of video. Based on the two-stream architecture, Wang et al. (2016) proposed a temporal segment network (TSN), which divided video sequences at fixed intervals to enhance the recognition ability of key actions through chronological segmented sampling and a sparse sampling strategy. Subsequently, Feichtenhofer et al. (2019) proposed SlowFast with two-branch structure, where slow branch receives images at a lower frame rate to extract spatial details, and another fast branch processes at a higher rate to capture motion cues. However, the two-stream network requires additional calculation of optical flow information, and it susceptible to illumination variations and occlusions, thus impacting action recognition accuracy. To sum up, the CNN-based action recognition methods are difficult to capture the complex temporal features of video, due to the limited receptive field of convolutional operations.

### 2.2 TRANSFORMER-BASED ACTION RECOGNITION

Unlike the localized receptive field of convolution, transformer architecture can achieve global information associations of entire videos by self-attention mechanism, facilitating extracting spatiotemporal features of action. Bertasius et al. (2021) presented a spatiotemporal transformer with the convolution-free architecture, which employed distinct spatial and temporal self-attention mechanisms to capture local relationship between adjacent patches and global dependency of video sequences. A multi-scale visual transformer architecture (Fan et al., 2021) was introduced, utilizing attention mechanism to capture visual and complex temporal information. Arnab et al. (2021) proposed a video vision transformer (ViViT), utilizing multiple transformer layers to obtain spatiotemporal information. Zhang et al. (2021) designed a zero-parameter token shift module to acquire temporal relationship of actions.

Afterwards, Liu et al. (2022) presented a hierarchical video swin transformer to extract video features in non-overlapping local windows.Yan et al. (2022) proposed multiview transformers (MTV), which used different encoder designs to capture multiple views of video actions. Chen et al. (2022) combined mobileNet and transformer architecture, and proposed Mobile-Former model to establish

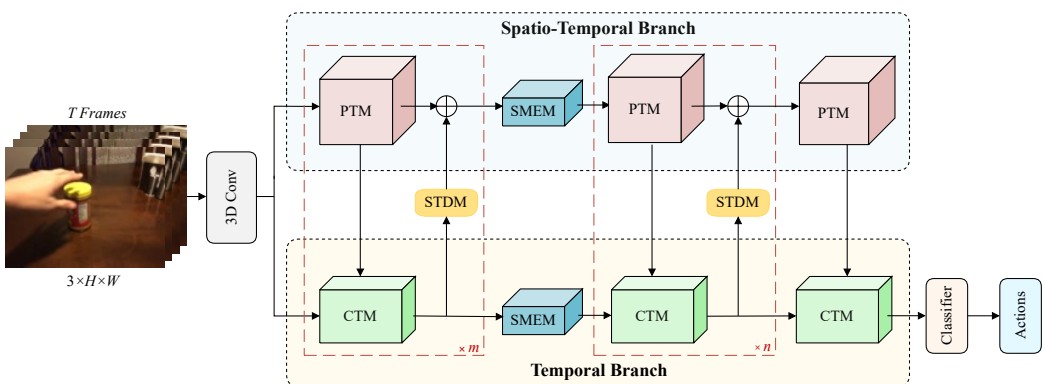

Figure 1: Overall architecture of the proposed approach.

a bidirectional bridge by cross-attention. Li et al. (2022) introduced the shrinking temporal attention Transformer (STAT), which efficiently builds spatiotemporal attention maps considering the attenuation of spatial attention in short and long temporal sequences. (Li et al., 2023) proposed UniFormer, which integrated convolution with spatiotemporal attention to effectively represent local details and capture global dependency relationship, achieving accurate recognition of actions. Venkataramanan et al. (2024) developed a DoRA model to capture key action features. Lee et al. (2023) constructed a cross-spatiotemporal attention module and proposed a new network based on transformer to enhance video spatiotemporal comprehension. Yang et al. (2023) proposed adapting image models (AIM), which leveraged transfer learning to transfer the pre-training image models to video models. GSoANet (Wang et al., 2023b) employed group second-order aggregation to enhance video action recognition. Sun et al. (2023a) proposed a windows and linear transformer (WLiT) that efficiently recognizes actions in videos by combining spatial-windows attention with linear attention. Lee et al. (2023) constructed a cross-spatiotemporal attention module and proposed a new network based on transformer to enhance video spatiotemporal comprehension. A self-supervised learning method, called multi-view videos (Shah et al., 2024), utilized a masked auto encoders (MAE) framework and enhanced the robustness of model by cross-view reconstruction. M2-CLIP (Wang et al., 2024) introduced multimodal adapters and a multi-task decoder for video action recognition. Xian et al. (2024) presented a new approach for aerial video action recognition through using mutual information to align temporal features and sampling. However, most action recognition approaches based on transformer ignore the extraction of local feature, and the high computational complexity of the self-attention mechanism reduces the inference speed of the model.

## 3 METHOD

### 3.1 OVERVIEW

Considering the challenge of long time spans of actions and the indistinguishable fine-grained features, we integrate transformer architecture mining global features with convolutional operator that can extract local information effectively, and propose a spatio-temporal diffusion transformer (STD-Former) model. STD-Former employs a dual-branch structure to capture the global spatiotemporal information in videos.

Figure 1 shows the architecture of STD-Former, which consists of the spatiotemporal and temporal branches respectively.Firstly, the videos are divided into the video sequence by sparse sampling strategy at fixed intervals. Subsequently, a 3D convolution module is employed to extract the feature of video sequence. Then, they are fed into a spatiotemporal branch and a temporal branch respectively. The spatiotemporal branch mainly contains twelve parallel transformer module (PTM), which parallelly extracts the temporal and spatial features of video by attention mechanism and convolutional structure respectively. Meanwhile, another temporal branch is mainly composed of twelve cross transformer module (CTM) (where $m + n = 11$ in Figure 1) to capture the temporal depen-

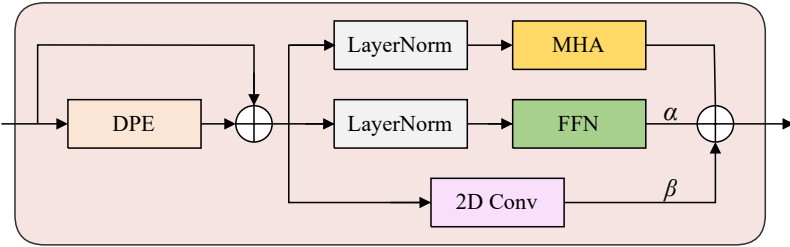

Figure 2: The structure of parallel transformer module.

dency relationship of actions, thus improving the temporal modeling capability. What's more, we leverage diffusion principle to mine short-term feature dependency, and design a spatio-temporal diffusion module (STDM) to transfer information from the temporal branch to the spatiotemporal branch, thereby enhancing the spatiotemporal feature. Furthermore, a lightweight salient motion excitation module (SMEM) that converts the spatial information from adjacent video frames into motion features is inserted in the two branches to enhance the feature representation for the fine-grained actions. Finally, the output feature from the last CTM module in the temporal branch is sent to the classifier to produce the action recognition result.

### 3.2 PARALLEL TRANSFORMER MODULE

The extraction of spatiotemporal information in videos is crucial to action recognition. We design a PTM to capture global spatiotemporal features of videos in this paper. Figure 2 illustrates the architecture of PTM which integrates a modified multi-head attention mechanism (MHA), a feed-forward neural network (FFN), and a two-dimensional convolutional layer (2D Conv) in parallel after dynamic position encoding (DPE).

Firstly, depthwise separable convolution based DPE encodes positional information of video actions, which is composed of a 1×1×1 convolution, a 3×1×1 convolution and a 1×1×1 convolution. Meanwhile, a residual connection is added to integrate the original video information. Then, the encoded features obtained by DPE are fed into the MHA, FFN and 2D convolution layer respectively. After layer normalization (LayerNorm), MHA and FFN are used to extract global spatiotemporal feature $y_1$ and $y_2$ through the weighted summation of multiple attention heads and the combination of linear transformation and activation function respectively. At the same time, we extract temporal feature $y_3$ from adjacent frames of videos through a 2D convolutional layer including a 1×1, a 3×3 and a 1×1 two-dimensional convolutional structure.

Finally, the video features acquired from MHA, FFN and 2D convolutional layer are fused. A feature fusion strategy with learnable parameters is developed to enhance the global spatiotemporal features of video actions.

$$y = y_1 + \alpha y_2 + \beta y_3, \tag{1}$$

where $y$ represents the output feature of PTM, $\alpha$ and $\beta$ are adjustment parameters that control the weights of features $y_2$ and $y_3$ respectively.

### 3.3 CROSS TRANSFORMER MODULE

To further explore the temporal relationship of actions among video frames, we design a CTM to enhance temporal features by fusing video semantics at different levels. The structure of CTM is shown in Figure 3, which mainly contains a convolutional position encoding (CPE), a cross multi-head attention (CMHA) and a FFN. CPE consists of a 3×3×3 3D convolution and dimension reshaping from $(T, H, W, C)$ to $(N, T, C)$, where $N = H \times W$. Then , cross-attention in CMHA is employed to realize the interaction of PTM and CTM from two branches after layer normalization, where the query matrix is derived from the current layer PTM, while the key and value matrices are sourced from the upper-layer CTM. Subsequently, the feature obtained by CMHA is processed through layer normalization and FFN to perform spatial transformation on video features. In addition, each mod-

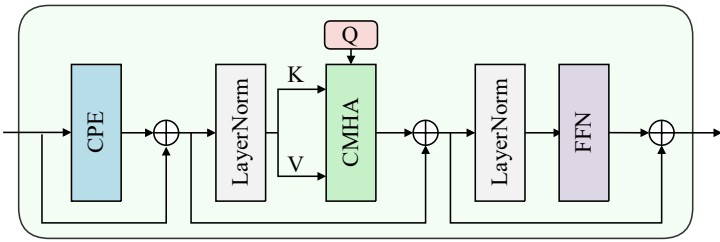

Figure 3: The structure of cross transformer module.

ule of CTM is following by a residual connection to avoid the loss of effective information, thus enhancing the temporal feature of video.

### 3.4 SPATIO-TEMPORAL DIFFUSION MODULE

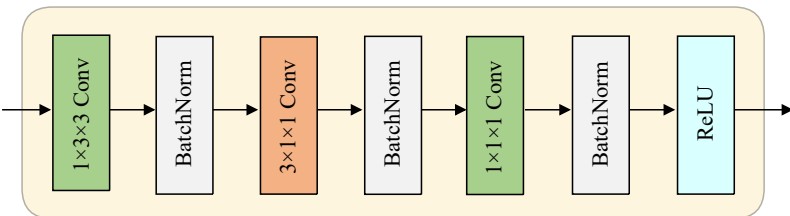

Figure 4: Structure of spatiotemporal diffusion module.

Inspired by the advantage of the diffusion principle for capturing long-distance relevant information, we propose a STDM to integrate temporal information of actions into the global representation of videos. Figure 4 depicts the structure of STDM , which consists of 1×3×3, 3×1×1, 1×1×1 convolution operators, batch normalization between adjacent conbolutions, and ReLU activation function. STDM learns local temporal relationships in video actions from the temporal branch features through a series of local convolution operations, and then diffuses them to the spatiotemporal branch, thereby accurately representing the long-term temporal dependency of actions.

In fact, STDM is plug-and-play, and can be flexibly integrated at any stage of the two-branch network. This module simulates the information propagation mechanism within the network, and continually passes the temporal features captured by the temporal branch to the spatiotemporal branch, thus enhancing the spatiotemporal feature of videos.

### 3.5 SALIENT MOTION EXCITATION MODULE

The change of the appearance features in videos can reflect the motion patterns of actions. To better extract video temporal information, we construct a lightweight SMEM, which converts spatial information from adjacent frames into motion features. Figure 5 shows the structure of SMEM. Initially, the input feature is process by a 3×3 2D convolution to extract local spatial information of the video. Next, to extract the feature correlation of video sequences, a correlation calculation module is introduced, which contains temporal feature separation, dimension reshaping, matrix multiplication and feature concatenation to enhance the features of adjacent video sequences and aggregate motion information at different moments. After the motion information is obtained, feature transformation is applied to enhance the temporal features of the video. This process consists of stacked 3×3 2D convolution layers, downsample operation, batch normalization (BatchNorm) and activation functions. Finally, a 3×3 2D convolution is utilized to further strengthen temporal information of the video, followed by an upsampling operation to restore the feature dimension. SMEM can capture the changes of action at different granularities from spatial features, thereby enhancing the feature representation of fine-grained actions.

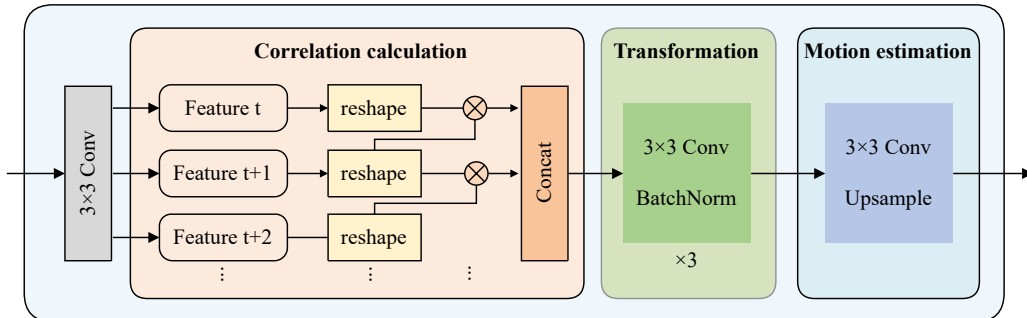

Figure 5: The structure of salient motion excitation module.

# 4 EXPERIMENTS

## 4.1 DATASET

Two mainstream datasets in the field of action recognition are used to verify the performance of the proposed STD-Former. Something Something V1 (SSV1) dataset is collected in real scenes and records the action information of different objects from multiple angles, including the subtle changes of the moving subject, which consists of 108,499 video clips. It describes the interactive actions between people and objects in different scenarios. These actions are divided into 174 categories(Raghav Goyal, 2017).

Another Something Something V2 (SSV2) dataset also has 174 action categories, but the number of videos has increased to 220,487(Raghav Goyal, 2017). This dataset not only records richer video action information but also greatly reduces the impact of label noise on action recognition. Different from other scene datasets that rely on video background for action prediction, there are very similar action scenes in SSV1 and SSV2 datasets, but they belong to different action categories, such as picking up and putting down objects, opening and closing containers, etc. Therefore, SSV1 and SSV2 belong to time-dependent datasets. They put forward higher requirements on the ability of action recognition methods to capture effective spatio-temporal information of videos. After that, random inversion operations are used to increase the number of model input samples.

## 4.2 EXPERIMENTAL SETTING

RGB video data of datasets are first sampled at an interval of 16 frames per second through a sparse time sampling strategy, and then are cropped to a size of 224×224. After that, random inversion operations are utilized to increase the number of model input samples. During training model, the initial learning rate and the training batch size (Batch Size) are set to 0.0001 and 16 respectively. The cosine annealing strategy is employed to update the current learning rate of the network. The common AdamW (Adam with Weight decay) and the soft cross-entropy function are used for the optimizer and the loss function of model respectively. In addition, STD-Former model is trained based on the parameters of Contrastive Language–Image Pre-training (CLIP). The model is executed on a NVIDIA RTX 4090 server, and the experimental results are obtained in PyTorch 1.10 framework in this paper.

## 4.3 COMPARISON WITH STATE-OF-THE-ART MODELS

To evaluate the performance of the proposed STD-Former, it is compared with some representative advanced action recognition methods, including the CNN-based models such as MSNet, TDN, TEA, CT-Net, AE-Net, and MSMA, as well as the transformer-based action recognition models such as TimeSformer, MViT, ViViT, MTV, AIM, and UniFormerV2. To ensure fairness in the comparison, all models utilize the same video sampling rate and testing strategy in the experiment.

Table 1: Comparison STD-Former and state-of-the-art methods on SSV1 and SSV2. 'IN-21K' and 'K400' represent pre-training by using ImageNet-21K and Kinetics-400 datasets respectively. The best results are marked in bold, and the second-best results are underlined. '-' represents that the information is not given.

| Methods | Pretrained | Input | SSV1 | | SSV2 | |
|---|---|---|---|---|---|---|
| | | | Top-1(%) | Top-5(%) | Top-1(%) | Top-5(%) |
| MSNet (Kwon et al., 2020) | ImageNet | 16×1×1 | 52.1 | 82.3 | 64.7 | 89.4 |
| TEA (Li et al., 2020) | ImageNet | 16×3×10 | 52.3 | 81.9 | 65.1 | 89.9 |
| TDN (Wang et al., 2021) | ImageNet | 24×1×1 | 55.1 | 82.9 | 67.0 | 90.3 |
| CT-Net (Li et al., 2021) | ImageNet | 16×3×2 | 53.4 | 81.7 | 65.9 | 90.1 |
| TimeSformer-L (Bertasius et al., 2021) | ImageNet | 64×1×3 | - | - | 62.4 | - |
| MSMA (Peng & Tseng, 2023) | ImageNet | 16×1×3 | 55.8 | 83.1 | 66.2 | 90.4 |
| AE-Net (Wang et al., 2023a) | ImageNet | 16×1×1 | 54.1 | 81.7 | - | - |
| MViT-B/16 (Fan et al., 2021) | K400 | 16×1×3 | - | - | 66.2 | 90.2 |
| ViViT-L/16×2 (Arnab et al., 2021) | IN-21K | 32×4×1 | - | - | 65.4 | 89.8 |
| MTV-B (Yan et al., 2022) | IN-21K | 32×4×3 | - | - | 67.6 | 90.4 |
| AIM-B/16 (Yang et al., 2023) | CLIP | 16×1×3 | - | - | 68.1 | 91.8 |
| UniFormerV2-B (Sun et al., 2023b) | CLIP-400M | 16×3×1 | 56.8 | 84.2 | **69.5** | **92.3** |
| STD-Former(Ours) | CLIP-400M | 16×3×1 | **57.3** | **84.4** | 69.2 | 92.1 |

Table 1 shows the experimental results of STD-Former and other advanced action recognition models on two temporal datasets SSV1 and SSV2. It can be seen from Table 1 that the Top-1 and Top-5 scores of STD-Former is highest than other methods for SSV1 dataset. The best performance of STD-Former on SSV1 dataset demonstrates that it can not only effectively extract the spatiotemporal features of long-term sequential actions but also capture fine-grained action information.

For SSV2 dataset, STD-Former is superior to other models except UniFormerV2-B in the Top-1 and Top-5 scores. The Top-1 and Top-5 accuracies of STD-Former are 0.3% and 0.2% lower than that of UniFormerV2-B, respectively. This is because STD-Former may ignore the influence of the complex background in videos. SSV2 dataset contains more and longer action videos than SSV1. However, the performance of STD-Former on SSV2 dataset is only slightly less than UniFormerV2-B, and far better than other action recognition methods except UniFormerV2-B in Table 1, which indicate that STD-Former can effectively model the long-term temporal dependence of actions and achieves competitive results.

### 4.4 ABLATION STUDY

The proposed STD-Former mainly is composed of our designed PTM, CTM STDM, and SMEM in this paper. To testify their effectiveness, a series of ablation experiments on SSV1 dataset are implemented to study their influence on the performance of STD-Former. Since CTM is essential for integrating the video features extracted by the spatiotemporal branch and the temporal branch, the CTM module is indispensable in the experimental process. So the model only containing CTM is regarded as the baseline model, where PTM is replaced by a conventional transformer module. The experimental results gotten by adding different modu les to the baseline model are shown in Table 2.

As can be seen from Table 2, the Top-1 and Top-5 accuracies of the baseline model are only 56.8% and 84.0%. After PTM is added into the baseline model, the Top-1 and Top-5 accuracies are observably increased to 57.2% and 84.3% respectively, which indicates that the global spatiotemporal features of video extracted by PTM is crucial to the action recognition. When the model concludes STDM and CTM, its Top-1 and Top-5 accuracies are 57.0% and 84.2% respectively, which is higher than that of the baseline, verifying the enhancement effect of STDM on spatiotemporal features. When SMEM and CTM are utilized simultaneously, the Top-1 and Top-5 accuracies of the model are slightly improved by 0.3% and 0.2% respectively. This shows that SMEM has certain advantages in identifying fine-grained actions. Finally, the Top-1 and Top-5 accuracies of STD-Former

Table 2: Component-wise Comparison in STD-Former on SSV1.

| PTM | STDM | SMEM | CTM | Top-1(%) | Top-5(%) |
|---|---|---|---|---|---|
| | | | ✓ | 56.8 | 84 |
| ✓ | | | ✓ | 57.2 | 84.3 |
| | ✓ | | ✓ | 57 | 84.2 |
| | | ✓ | ✓ | 57.1 | 84.2 |
| ✓ | ✓ | ✓ | ✓ | **57.3** | **84.4** |

are both improved, reaching 57.3% and 84.4% respectively, when all four modules are integrated. This result verifies the positive impact of the synergy among modules on improving the accuracy of action recognition. In summary, the constructed modules and the integration strategy of modules ensures favorable recognition ability of STD-Former for video actions in this paper.

## 4.5 STRATEGY ANALYSIS

**The design strategy of PTM**. PTM utilizes a 2D convolutional layer to extract temporal feature from adjacent frames of videos except the transformer architecture. In order to evaluate the structural rationality of PTM, a group of experiments on the placement and substitute of 2D convolution are carried out.

Table 3 shows the influence of different design strategies of PTM on the model. If 2D convolutional layer is placed in the back of the attention module, the Top-1 and Top-5 accuracies are 56.8% and 83.9% respectively. When a two-dimensional convolutional structure is added to the residual connection (that is PTM), the performance of the model is improved, reaching a Top-1 accuracy of 57.0% and a Top-5 accuracy of 84.1%, which shows that the strategy of adding 2D convolutional structure into the residual connection can achieve the best performance. Additionally, if 2D convolution is replaced by 3D convolution, the strategy of adding 3D convolution causes lower performance no matter where 3D convolution is placed. It indicates that the excessive usage of 3D convolution may bring in redundant information, thus reducing the recognition accuracy.

Table 3: The influence of different design strategies of PTM. 'Attention' represents the attention module. 'Residual' represents the residual connection. '2D Conv' and '3D Conv' represent the temporal information obtained by 2D convolution and 3D convolution respectively.

| Design strategies | Top-1(%) | Top-5(%) |
|---|---|---|
| Attention + 2D Conv | 56.8 | 83.9 |
| Attention + 3D Conv | 55.6 | 82.9 |
| Residual + 2D Conv | **57.2** | **84.3** |
| Residual + 3D Conv | 54.5 | 81.8 |

**The fusion strategy of SMEM**. Different fusion ways affect the integration of effective information. SMEM fuses temporal information by multiplication in the correlation calculation. To explore the impact of different fusion patterns on the accuracy of the action recognition model, SMEM with different fusion strategies are integrated into the baseline only containing CTM respectively, and the corresponding experimental results are shown in Table 4.

As can be seen from Table 4, the fusion strategy based on multiplication achieves a Top-1 accuracy rate of 57.1% and a Top-5 accuracy rate of 84.2%, which has the best performance among the three strategies. Even if the multiplication and addition strategies are employed concurrently, its performance is still lower than the model based on multiplication, which shows that the superposition of similar information may reduce the focus on key information. Therefore, SMEM employs the multiplication strategy to fuse video features and improve the overall recognition accuracy of the model.

Table 4: The influence of different fusion strategies of SMEM.

| Design strategies | Top-1(%) | Top-5(%) |
|---|---|---|
| Multiplicative | **57.1** | **84.2** |
| Addition | 56.9 | 84.0 |
| Multiplicative + Addition | 57.0 | 84.2 |

## 5 CONCLUSION

In this paper, we propose a STD-Former model to accurately identify the fine-grained and long time span actions from highly similar video backgrounds. STD-Former draws on the advantage of transformer architecture in modeling global information and a dual-branch structure is established, including a spatiotemporal branch and a temporal branch. Firstly, PTM is designed to mine the spatiotemporal information of videos by a parallel structure, and a two-dimensional convolution layer is added to enhance the temporal features of actions in the spatiotemporal branch. Meanwhile, in the temporal branch, CTM is created to extract the long-distance temporal dependencies of video actions. Furthermore, a novel STDM is presented to gradually feed back the temporal information of temporal branch to the spatiotemporal branch, thus enhancing the spatiotemporal features through inter-branch information diffusion. Additionally, to capture the fine-grained action features between adjacent video sequences, a lightweight SMEM is constructed and embedded into the two branches. Finally, the extracted spatiotemporal and motion features from both branches are fused to produce action prediction. The experimental results on SSV1 and SSV2 datasets show that STD-Former has favorable performance for the long-distance and fine-grained actions recognition.

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

APPENDIX

## A STRATEGY ANALYSIS

**The deployment of PTM**. In the experiment, we found that the different deployments of PTM can affect the performance of the action recognition model. A series of experiments are implemented on SSV1 dataset to discuss the best location to deploy PTM. As mentioned in Section 3.1, the spatiotemporal branch of STD-Former contains twelve attention modules (sequentially from 0 to 11), and PTM can be placed in any one or more locations on the branch. For the locations where PTM is not deployed, the conventional transformer module is used as an alternative. The experimental results of the model only including PTM and CTM are shown in Table 5.

We first deploy PTM at each location of the spatiotemporal branch in STD-Former (0,1,2,3,4,5,6,7,8,9,10,11), and then gradually reduce the number of PTM. As can be seen from Table 5, when PTM is deployed in the last four positions (8,9,10,11) of the spatiotemporal branch, Top-1 and Top-5 accuracies of the model reach 57.2% and 84.1% respectively, which is the best performance among all tested locations. This result indicates that when the PTM module is deployed in the last one-third of the spatiotemporal branch, STD-Former can effectively integrate the deep semantic information of videos and capture the spatiotemporal features of actions, thereby improving the overall recognition accuracy of the model.

Table 5: The influence of different locations of PTM.

| the deployment position of the PTM module | Top-1(%) | Top-5(%) |
|---|---|---|
| {0,1,2,3,4,5,6,7,8,9,10,11} | 55.2 | 82.6 |
| {0,1,2,3} | 55.8 | 82.9 |
| {4,5,6,7,8,9,10,11} | 56.6 | 83.5 |
| {4,5,6,7} | 57.0 | 84.1 |
| {8,9,10,11} | **57.2** | **84.3** |
| {8,9} | 56.8 | 84.0 |
| {10,11} | 56.6 | 83.4 |

**The joint deployment of PTM and SMEM**. To explore the impact of different combinations of PTM and SMEM on the model performance, a group of experiments are carried out. Similar with the previous subsection, PTM is deployed in any one or more locations (sequentially from 0 to 11) on the spatiotemporal branch in STD-Former, and at the positions where PTM is not deployed, PTM is replaced by the conventional transformer module. The plug-and-play SMEM can be inserted into any position (sequentially from 0 to 11) of the spatiotemporal or temporal branch. Table 6 shows the experimental results of STD-Former with different combinations of PTM and SMEM.

From the experimental results in Table 4, we have found that the model has better performance when PTM is deployed in the middle ({4,5,6,7}) and last four ({8,9,10,11}) positions of the spatiotemporal branch. On this basis, SMEM is deployed at different locations of the spatiotemporal (ST-branch) or temporal branch (T-branch). Experiments found that the accuracy of the model is low if SMEM is inserted into the spatiotemporal and temporal branches simultaneously, so the corresponding results are not listed. It can be seen from Table 6 that when PTM is deployed in the last four positions of the spatiotemporal branch ({8,9,10,11}) and SMEM is inserted into the fifth position of the temporal branch (T-branch {4}), STD-Former has the highest Top-1 and Top-5 accuracies, reaching 57.3% and 84.4% respectively. It manifests that the shallow feature of the temporal branch contains abundant motion information, so the performance of the model can be further improved by using SMEM to extract key motion features from adjacent video sequences.

Table 6: The combination of PTM and SMEM module.

| PTM | SEME | Top-1(%) | Top-5(%) |
|---|---|---|---|
| {4,5,6,7} | ST-branch {3} | 56.9 | 84.0 |
| {4,5,6,7} | ST-branch {4} | 57.0 | 84.0 |
| {4,5,6,7} | ST-branch {7} | 56.6 | 83.7 |
| {8,9,10,11} | T-branch{3} | 57.2 | 84.2 |
| {8,9,10,11} | T-branch{4} | **57.3** | **84.4** |
| {8,9,10,11} | T-branch{7} | 57.0 | 84.1 |

## B  PARAMETERS ANALYSIS

The proposed PTM integrates the features by two adjustment parameters, namely $\alpha$ and $\beta$ in formula (1). Table 7 shows the experimental results of STD-Former on SSV1 dataset under different adjustment parameters $\alpha$ and $\beta$.

As can be seen from Table 7, when the values of parameters $\alpha$ and $\beta$ are both 1.0, STD-Former achieves the optimal recognition accuracy, which means that the features obtained by attention mechanism and convolutional structure are equally important. Whether the parameters $\alpha$ and $\beta$ are increased or decreased, the performance of STD-Former slowly declines. It demonstrates that both the attention mechanism and convolutional structure in PTM can effectively extract spatiotemporal information of video, thus improving the performance of STD-Former.

Table 7: The different parameters of PTM module.

| $\alpha$ | $\beta$ | Top-1(%) | Top-5(%) |
|---|---|---|---|
| 0.5 | 0.5 | 55.6 | 83.0 |
| 0.8 | 0.8 | 56.9 | 83.9 |
| 1.0 | 1.0 | **57.3** | **84.4** |
| 1.2 | 1.2 | 57.0 | 84.0 |
| 1.5 | 1.5 | 56.8 | 83.7 |

