# OpenReview forum: "Spatio-temporal Diffusion Transformer for Action Recognition"
_ICLR.cc/2025/Conference — Submitted to ICLR 2025_

### Official Review · Reviewer_Htwf · 2024-10-28

**Soundness:** 2
**Presentation:** 2
**Contribution:** 1
**Rating:** 3
**Confidence:** 5

**Summary:**

This paper introduces a dual-path network consisting of a spatio-temporal pathway and a temporal pathway. The authors also propose a series of network blocks designed to enhance the action modeling capability. However, these proposed blocks lack novelty.

**Strengths:**

1. The writing is neat.

**Weaknesses:**

1. Outdated Paradigm: The approach of proposing spatial-temporal network blocks and small models specific to the action recognition problem to achieve state-of-the-art (SOTA) performance is an outdated paradigm. This approach was popular before 2021, such as TDN[1]. Modern paradigms focus on developing general visual models, such as CLIP and large multi-modal models, and transferring their visual representations to action recognition tasks, by parameter-efficient tuning (PET). Representative works in this area include  ST-Adapter[2].

2. Lack of Novelty in Network Blocks: The proposed network blocks are not novel. The mixed convolution + Attention block (shown in Fig. 2) was proposed in [3]. The cross transformer module (Fig. 3) was introduced in [4]. The spatiotemporal diffusion module (Fig. 4), which is essentially a factorized 3D convolution, was proposed in [5]. The motion excitation module (Fig. 5) was introduced in [6][7].

3. Poor Performance: The performance of the proposed model is subpar.

[1]Wang L, Tong Z, Ji B, et al. Tdn: Temporal difference networks for efficient action recognition[C]//Proceedings of the IEEE/CVF conference on computer vision and pattern recognition. 2021: 1895-1904.
[2]Pan J, Lin Z, Zhu X, et al. St-adapter: Parameter-efficient image-to-video transfer learning[J]. Advances in Neural Information Processing Systems, 2022, 35: 26462-26477.
[3]Pan X, Ge C, Lu R, et al. On the integration of self-attention and convolution[C]//Proceedings of the IEEE/CVF conference on computer vision and pattern recognition. 2022: 815-825.
[4]Chen C F R, Fan Q, Panda R. Crossvit: Cross-attention multi-scale vision transformer for image classification[C]//Proceedings of the IEEE/CVF international conference on computer vision. 2021: 357-366.
[5]Tran D, Wang H, Torresani L, et al. A closer look at spatiotemporal convolutions for action recognition[C]//Proceedings of the IEEE conference on Computer Vision and Pattern Recognition. 2018: 6450-6459.
[6]Wang H, Tran D, Torresani L, et al. Video modeling with correlation networks[C]//Proceedings of the IEEE/CVF conference on computer vision and pattern recognition. 2020: 352-361.
[7]Li Y, Ji B, Shi X, et al. Tea: Temporal excitation and aggregation for action recognition[C]//Proceedings of the IEEE/CVF conference on computer vision and pattern recognition. 2020: 909-918.

**Questions:**

1. Why stick to the old "network block" game when modern paradigms are available? Specialized small models are outdated, and general models are the future.

2. clearly highlight any unique contributions or improvements over existing methods to establish the novelty of the proposed modules.

3. Why not benchmark the model on more challenging video datasets such as Kinetics700?

---

### Official Review · Reviewer_7sTy · 2024-11-03

**Soundness:** 1
**Presentation:** 1
**Contribution:** 2
**Rating:** 3
**Confidence:** 5

**Summary:**

This paper proposes a method for video action recognition by designing a two-transformer-branch for extracting spatio-temporal features. Furthermore, it applies a spatio-temporal diffusion model to fuse the features from the two branches for enhancing the video features. To better process temporal information, it constructs a salient motion extraction module to convert spatial information from adjacent frames into motion features. Experiments are conducted on Something Something V1 and V2.

**Strengths:**

1. The idea is simple and straightforward.
2. Applying diffusion model for action recognition is a new try.

**Weaknesses:**

1. Overall, paper writing is poor. The motivation of the proposed method is not clear. Experiments are not enough to support the proposed method.

2. Abstract:
“the high similarity of video backgrounds and the long time span of action bring serious challenges to action recognition”. How did existing methods handle the issue? What shortages of existing methods on this issue? Why the proposed method is able to solve this issue?

3. Introduction:
The introduction is more like a Related Work. There is no motivation introduced. From L82, it generally describes the proposed method, but  it is not clear why the method is designed step by step like this? How the design to solve the issue mentioned above in the Abstract?

4. Method:
1) It mentions that one brach is for extracting spatiotemporal features for video, another is for temporal feature of actions. What difference between video features and action features here? Why extra temporal features needed? Are there any redundancy between the spatio-temporal features and temporal features? Is there any ablation study here?
2) Why utilize 12 modules in PTM? Is it a hype-parameter? Is there any ablation study here?
3) It is not clear that how diffusion mode works on spatio-temporal dimension here? What problem it solves?
4) It difficult for me to understand the SMEM? What is the motivation to design the model? How important it is for the propose method?

5. Experiment:
1) From Table 2, results of each module are not significant different. The table does not well support that the proposed modules are necessary.
2) Same with results from Table 4. There is no significant difference between different settings.
3) Experiments on only Something-Something are not enough to prove the effectiveness and generalizability.

6. From Table 2, the diffusion model does not make big difference. And also it is a very small part of the method. I don't understand why the title is spatio-temporal diffusion Transformer. How spatio-temporal diffusion contribution here?

7. All figures and tables miss detailed captions. It makes reading difficult.

**Questions:**

See the above Weaknesses.

---

### Official Review · Reviewer_7geJ · 2024-11-04

**Soundness:** 2
**Presentation:** 2
**Contribution:** 1
**Rating:** 3
**Confidence:** 4

**Summary:**

This paper proposes a spatio-temporal diffusion(really?) transformer to improve action recognition task. Several plug-and-play modules are proposed, including Parallel Transformer Module, Cross Transformer Module, Spatiotemporal Diffusion(really?) Module and Motion Excitation Module.

**Strengths:**

The experiments on Something Something V1 and V2 datasets show the effectiveness of proposed modules.

**Weaknesses:**

- This paper is titled with "Spatio-temporal Diffusion Transformer", but I don't see any context about "Diffusion". I am not sure this is a typo(for example, "Fusion" may be more appropriate) or the authors misuse the concept of Diffusion.
- Utilizing spatiotemporal and temporal information to improve video tasks is a well-known knowledge in this field. But I don't learn any new idea from this work, either from research insights or engineering implementations.

**Questions:**

I really suggest authors to double-check the concept of Diffusion.

---

### Meta-Review · Area_Chair_fNVd · 2024-12-24

**Metareview:**

This paper introduces a dual-path network consisting of a spatio-temporal pathway and a temporal pathway for action recognition. Although the experiments on Something Something V1 and V2 datasets show the effectiveness of proposed modules, all three reviewers point that the proposed method is lack of novelty and rate this paper as "reject". After reading the paper, the AC agrees with the reviewers' comments about novelty.

**Additional Comments On Reviewer Discussion:**

The authors do not provide feedback for the reviewers' comments.

---

### Decision · Program_Chairs · 2025-01-22

Reject